# The Barrier-Enhancing Function of Soluble Yam (*Dioscorea opposita* Thunb.) Polysaccharides in Rat Intestinal Epithelial Cells as Affected by the Covalent Se Conjugation

**DOI:** 10.3390/nu14193950

**Published:** 2022-09-23

**Authors:** Zhen-Xing Wang, Xin-Huai Zhao

**Affiliations:** 1Key Laboratory of Dairy Science, Ministry of Education, Northeast Agricultural University, Harbin 150030, China; 2School of Biology and Food Engineering, Guangdong University of Petrochemical Technology, Maoming 525000, China; 3Research Centre of Food Nutrition and Human Healthcare, Guangdong University of Petrochemical Technology, Maoming 525000, China; 4Maoming Branch, Guangdong Laboratory for Lingnan Modern Agriculture, Guangdong University of Petrochemical Technology, Maoming 525000, China

**Keywords:** barrier function, intestinal epithelial cells, ROCK/RhoA pathway, selenylation, yam polysaccharides

## Abstract

The non-starch yam polysaccharides (YP) are the bioactive substances of edible yam, while Se is an essential nutrient for the human body. Whether a covalent conjugation of Se to YP might cause bioactivity change for the resultant selenylated YP in the intestine is still insufficiently studied, including the critical intestinal barrier function. In this study, two selenylated YP products, namely, YPSe-I and YPSe-II, with corresponding Se contents of 795 and 1480 mg/kg, were obtained by the reaction of YP and Na_2_SeO_3_ in the presence of HNO_3_ and then assessed for their bioactivities to a cell model (i.e., rat intestinal epithelial IEC-6 cells). The results showed that YP, YPSe-I, and YPSe-II at 5–80 μg/mL dosages could promote cell growth with treatment times of 12–24 h. The three samples also could improve barrier integrity via increasing cell monolayer resistance and anti-bacterial activity against *E. coli* or by reducing paracellular permeability and bacterial translocation. Additionally, the three samples enhanced F-actin distribution and promoted the expression of the three tight junction proteins, namely, zonula occluden-1, occludin, and claudin-1. Meanwhile, the expression levels of ROCK and RhoA, two critical proteins in the ROCK/RhoA singling pathway, were down-regulated by these samples. Collectively, YPSe-I and, especially, YPSe-II were more potent than YP in enhancing the assessed bioactivities. It is thus concluded that this chemical selenylation of YP brought about enhanced activity in the cells to promote barrier integrity, while a higher selenylation extent of the selenylated YP induced much activity enhancement. Collectively, the results highlighted the important role of the non-metal nutrient Se in the modified polysaccharides.

## 1. Introduction

Trace elements such as Cr, Cu, Fe, Mn, Zn, Se, and others play a particular and important role in the body, because they are involved in various physiological processes including cell metabolism, immune regulation, bone formation, and acid-base balance regulation and are also the essential or auxiliary components of many hormones and enzymes [1,2,3]. Generally, an excessive or insufficient dietary intake of these trace elements can cause the occurrence of various related diseases and even fatal hazards [4,5]. For the non-metal nutrient Se, it is well known that Se is an essential component of these enzymes, such as glutathione peroxidase, thioredoxin reductase, and iodothyronine deiodinase, in the body [6,7,8], and more importantly, these enzymes play a crucial role in protecting cellular lipids, lipoproteins, and DNA from oxidative damage [9]. Natural foods generally are lower in Se content; meanwhile, Se is only available through the dietary approach. Thus, malnutrition due to Se deficiency has become a problem in some regions. It is agreed that the organic Se is less toxic, more bioactive, and safer in the organism than the inorganic Se [10,11]. Thus, a conversion of the inorganic Se into the organic Se via various chemical reactions and microorganisms had been investigated. For example, both proteins and polysaccharides could be selenylated [12,13], while yeast could be employed to convert the inorganic Se into the organic Se [14]. Subsequently, the Se-enriched yeast was found to be able to improve the cognitive decline in the model mice and reduce the Aβ and tua lesions in the brain tissue [15]. The researchers thus pay attention to the bioactivities of the selenylated protein/polysaccharide products using both cell and animal models.

It is worth mentioning that the bioactivities of natural proteins and polysaccharides can be efficiently altered after the covalent Se binding. For example, the selenylated ovalbumin was observed with higher anti-oxidation [13]. In general, the anti-oxidant and hypoglycemic effects are two important biofunctions of polysaccharides. It had been revealed that the selenylated polysaccharides from *Sargassum pallidum* were more effective than the non-selenylated counterparts in scavenging both 2,2′-azino-bis(3-ethylbenzothiazoline-6-sulfonic acid) and 1,1-diphenyl-2-picrylhydrazyl (ABTS and DPPH) radicals [16], while the Se-polysaccharides isolated from *Cordyceps sinensis* had enhanced the anti-oxidant capacity in the streptozotocin-induced diabetic rats and could accelerate the binding of blood glucose to serum albumin to decrease the blood glucose level [17]. In addition, polysaccharides are regarded as possessing anti-tumor and immune-regulation effects. For example, the Se-enriched *Grifola frondosa* polysaccharide could enhance the thymus and spleen indices of the tumor-bearing mice, increase both serum tumor necrosis factor-α (TNF-α) and interleukin-2 (IL-2) levels, and also promote nitric oxide (NO) production and macrophage phagocytosis [18]. Furthermore, natural polysaccharides have been assessed for their capacity to protect the intestinal barrier. The results declared that the *Cordyceps sinensis* polysaccharides could increase the intestinal villus length and crypt depth and promote the goblet cells and mucin expression in the cyclophosphamide-injured mice [19]. Meanwhile, the *Atractylodes macrocephala* polysaccharides were capable of attenuating the damaging effect of dextran sulfate sodium (DSS) on small intestinal epithelial cells, thus preventing the decreased expression of these tight junction (TJ) proteins such as occludin, claudin-1, and zonula occludin-1 (ZO-1) [20]. However, whether the selenylated polysaccharides have increased or decreased barrier promotion in the intestine remains less assessed so far.

In the previous studies of our group, it was found that the soluble and non-digestible yam (*Dioscorea opposita* Thunb.) polysaccharides (YP) had immune-regulation [21], while a chemical Se conjugation of YP led to enhanced immune-regulation [22]. Thus, compared with the unmodified YP, whether the selenylated YP (YPSe) also possessed helpful enhancement in their barrier promotion deserves a detailed investigation. In this study, the water-soluble and non-starch YP were thus extracted from fresh yam and then selenylated chemically to obtain two selenylated YP products (i.e., YPSe-I and YPSe-II), utilizing a covalent chemical selenylation mediated by Na_2_SeO_3_-HNO_3_. Afterwards, the bioactivities of the prepared YP, YPSe-I, and YPSe-II to the rat intestinal epithelial (IEC-6) cells were assessed and compared for these critical barrier-related indices, including cell viability, transepithelial resistance (TEER), paracellular permeability, anti-bacterial activity, bacterial translocation, the intercellular distribution of filamentous actin (F-actin), as well as the expression of several TJ proteins. Moreover, the expression of several genes and proteins of the cells that are critical to the ROCK/RhoA signaling pathway were clarified to support the barrier promotion of the three polysaccharide samples. This study aimed to reveal the possible barrier promotion of YP in the intestine and, more importantly, to identify how a chemical selenylation of YP would bring about a changed YP activity to the intestinal epithelial cells regarding their important barrier integrity.

## 2. Materials and Methods

### 2.1. Regents and Materials

The fresh yam, cultured in Wenxian County (Henan Province, China), was purchased from the local market in Harbin (Heilongjiang Province, China). The fluorescein sodium salt (FS-Na), 4 kDa fluorescein isothiocyanate-dextran (FD-4), 3-(4,5-dimethyl-2-thiazolyl)-2,5-diphenyl tetrazolium bromide (MTT), and Dulbecco’s modified Eagle’s medium (DMEM) were purchased from Sigma-Aldrich Co. (St. Louis, MO, USA). The phosphate-buffered saline (PBS), α-amylase, Luria Bertani (LB) agar, phosphate-buffered saline (PBS), and dimethyl sulfoxide (DMSO) were bought from Solarbio Science and Technology Co., Ltd. (Beijing, China). The alkaline protease was bought from Beijing Aoboxing Biotechnologies, Inc. (Beijing, China), while the radio-immunoprecipitation assay (RIPA), phenylmethanesulfonylfluoride (PMSF), Actin-Tracker Red-Rhodamine, and bicinchoninic acid (BCA) kits were provided by the Beyotime Institute of Biotechnology (Shanghai, China). Other chemicals of analytical grade were employed in this study. The water was ultrapure water generated from a Milli-Q Plus (Milipore Corporation, New York, NY, USA).

RNAprep Pure Cell Bacteria Kit was purchased from Tiangen Biotech Co., Ltd. (Beijing, China), while the NovoScript^®^ Two-Step RT-PCR Kit and SYBR qPCR SuperMix Plus were purchased from Novoprotein Biotech Co., Ltd. (Suzhou, China). The primary antibodies (GAPDH Bioss bs10900R, ZO-1 Bioss bs-1329R) were purchased from Bioss Biotechnology Co., Ltd. (Beijing, China). The claudin-1 (28674-1-AP) and occludin (13409-1-AP) were purchased from Proteintech Group Inc. (Wuhan, China), while ROCK1 (#4035) and RhoA (#2117) were obtained from Cell Signaling Technology (Danvers, MA, USA). Goat anti-rabbit secondary antibody was bought from Bioss Biotechnology Co., Ltd. (Beijing, China). The bacterial strain *Escherichia coli* (*E. coli*) ATCC 25,922 was provided by the Food Microbiology Laboratory in Northeast Agricultural University (Harbin, China), which was stored at −80 °C until use.

### 2.2. Cell Line and Cell Culture

IEC-6 cells were acquired from the American Type Culture Collection (Rockville, MD, USA). The cells were cultured in the DMEM containing 10% fetal bovine serum, 1% sodium pyruvate, 0.1% units/mL bovine insulin, and 100 U/mL penicillin/streptomycin and afterward mainlined at 37 °C in a humidified environment containing 5% CO_2_.

### 2.3. Preparation and Chemical Selenylation of YP

The extraction of YP was performed utilizing the strategy given in a past report [21], with a minor change. Briefly, the fresh yam was peeled, cut into small pieces, and mashed by a high-speed blender for 5 min. After that, the mashed yam was blended with distilled water at a fixed ratio of 1:10 (*w*/*w*), added with α-amylase (20 U/mL), and kept at 85 °C for 4 h with gentle stirring. The supernatant was collected by centrifugation at 8000× *g* for 15 min and concentrated at 100 °C to half of the original volume. The concentrated supernatant was digested by the alkaline protease alcalase (100 U/mL) at 55 °C for 6 h, heated at 100 °C for 5 min to inactivate the alcalase, centrifuged at 8000× *g* for 5 min, and concentrated again at 100 °C to reach one-fifth of the original volume. Anhydrous ethanol was added to the resultant supernatant to obtain a final ethanol concentration of 80% (*v*/*v*) and placed at 4 °C for 24 h. The sediment substances collected were YP, which were freeze-dried and then stored at −20 °C before use.

The chemical selenylation of YP followed a reported method [23]. Briefly, 1 g YP was dissolved in 10 mL of HNO_3_ (5%, *v*/*v*), added with 50 or 100 mg Na_2_SeO_3_, and heated at 75 °C for 8 h under a gentle stirring. After the chemical selenylation, the reaction mixture was cooled to 20 °C and added with absolute ethanol (five-times volume) to remove the unreacted H_2_SeO_3_, while the resultant sediment substances were YPSe-I and YPSe-II, respectively. The two selenylated products were washed with absolute ethanol, subjected to freeze-drying, and stored at −20 °C before use.

### 2.4. Measurements of Se and Saccharide Contents

An inductively coupled plasma-mass spectrometer (Agilent Technologies, Santa Clara, CA, USA) was used to quantify the Se content, as previously described [24]. Additionally, glucose was used as the standard to measure the total saccharide content using the phenol-H_2_SO_4_ method [25].

### 2.5. Evaluation of Cell Viability and Na_2_SeO_3_ Cytotoxicity

As previously mentioned, the viability value of the cells was evaluated using the MTT colorimetric method [26]. In detail, the cells of 2 × 10^3^ were seeded into 96-well plates and then serum-starved for 12 h after adherence for 24 h. Afterwards, the cells were treated with 100 µL of three samples (dose levels of 5–80 µg/mL) for 12 and 24 h. After discarding the supernatants, MTT (0.5 µg/mL) of 100 µL was added into each well, while the cells were cultured for an additional 4 h at 37 °C. After removing the supernatants, each well received 100 µL of DMSO, while the cells underwent a second 2 h incubation at 37 °C. The optical density of each well at 490 nm was then measured by a microplate reader (Bio-Rad Laboratories, Hercules, CA, USA). The viability value was calculated as previously described [27]. The cells treated with the culture medium only were used as the negative control, with a designed cell viability value of 100%.

To determine the cytotoxic effect of Na_2_SeO_3,_ the cells were cultured in a serum-free medium for an additional 12 h after being injected in 96-well plates with a regular medium for 24 h, The cells were washed with PBS (10 mmol/L, pH 7.2), while Na_2_SeO_3_ was added at final doses of 0.013, 0.13, and 1.3 μg/mL to treat the cells for 12 and 24 h. The MTT method was then used to measure the viability value, as above.

### 2.6. Assays of TEER and Paracellular Permeability

The transwell inserts (12 mm in diameter, 0.4 μm pore size, polyester membranes, Corning) were injected with a cell suspension of 0.5 mL (4 × 10^6^ cells/mL), and the basolateral side was then added with the culture medium of 1.5 mL. Every other day, the culture media were replaced, and the Millicell-ERS2 volt ohmmeter (Millipore, Bedford, MA, USA) was used to measure the TEER value. Until the detected TEER value reached 50 Ω cm^2^, the cells were serum-starved for 12 h. After that, the apical side was added with 0.5 mL DMEM medium containing YP, YPSe-I, and YPSe-II (final doses of 5–40 μg/mL), while the basolateral side was added with the culture medium of 1.5 mL. TEER change was then measured at 12 and 24 h. The cells treated with the culture medium were also designed as the negative control, while the TEER change was expressed as TEER (%) = (TEER_treatment_/TEER_control_) × 100 [28].

The paracellular permeability of the assessed cell monolayer was also measured in the Transwell inserts (12 mm diameter, 0.4 μm pore size, polyester membranes, Corning) using the FD-4 and FS-Na methods [29,30]. When the monolayer was cultured to a TEER value of 50 Ω cm^2^, FD-4 (0.5 mg/mL) or FS-Na (0.08 mg/mL) of 0.5 mL, together with the culture medium of 1.5 mL, was added to the apical and the basolateral sides, respectively. After an incubation at 37 °C for 24 h, the culture medium of 100 μL was removed from the basolateral side and measured for the fluorescence intensity using a fluorescent microplate reader (Infinite M200 pro, TECAN, Männedorf, Switzerland) and excitation/emission wavelengths of 490/520 nm. According to a previous study [31], the paracellular permeability was also expressed as a percentage value in comparison to the control cells.

### 2.7. Assays of Anti-Bacterial Activity and Bacterial Translocation

The cells were treated by these conditions employed in the TEER assay. Afterwards, the cell supernatants from different treatments were taken at time points of 12 and 24 h. The resistance of these supernatants to *E. coli* ATCC-25922 was evaluated as previously described [32]. Briefly, the bacterial solution (1 × 10^6^ CFU/mL) of 10 μL was mixed with one of the cell supernatants of 500 μL and cultured at 37 °C for 2 h. The combination, after being diluted with a 10-fold gradient, was then inoculated on the LB agar medium at 37 °C for 18 h, while the estimated colony counts were calculated. In this test, the cells exposed to the culture medium were served as the control.

The cells were inoculated into the Transwell inserts to reach the targeted TEER value of 50 Ω cm^2^, added with the culture medium containing the polysaccharide samples at a dose of 40 μg/mL, and then incubated for 12 and 24 h. The cells only treated with the culture medium were employed as the control. After that, *E. coli* (10^5^ CFU/well) was added to the apical side of the inserts. The basolateral compartments (100 μL) of the inserts were taken at the time points of 1–4 h and inoculated on the LB agar to detect the numbers of the *E. coli*-positive inserts as previously described [33].

### 2.8. Observation of Cytoskeletal F-Actin

A slightly modified version of the previously described [34] rhodamine-phalloidin labeling was used to observe the distribution of F-actin in the treated cells. In detail, the cells inoculated at 4 × 10^6^ cells/mL in 1 mL were placed in 12-well plates, treated with the samples at a dose of 40 µg/mL for 12 and 24 h, washed twice with the PBS (0.1 mol/L, pH 7.2), fixed with 4% paraformaldehyde for 15 min, and then washed twice with 0.1% Triton X-100 in PBS for 5 min. The Actin-Tracker Red-Rhodamine was diluted and applied to the fixed cells in accordance with the F-actin kit’s instructions. After incubation at 20 °C for 1 h in the dark, the cells were stained for the nuclei with 4′,6-diamidino-2-phenylindole (DAPI, 10 mg/L) of 500 μL for 5 min and then observed directly under a fluorescence microscope (OLYMPUS IX71, OLYMPUS Corporation, Tokyo, Japan) to obtain the features of cytoskeletal F-actin in the treated cells.

### 2.9. Quantitative Real-Time PCR Assay

The cells were plated into 6-well plates at a density of 3 × 10^5^ cells/well and exposed to the three polysaccharide samples (a dose of 40 µg/mL) for 24 h. The NovoScript^®^ two-step RT-PCR Kit was used to reverse transcribe the total RNA into complementary DNA (cDNA), while the NovoScript^®^ SYBR qPCR SuperMix Plus and Biosystems StepOnePlus real-time PCR system were used to amp up the cDNA. The total RNA was extracted in accordance with the instructions of the RNAprep Pure Cell Kit (Life Technologies Corporation, Carlsbad, CA, USA). The primers used in this experiment were provided by Sangon Biotech (Shanghai, China) Company Limited (Shanghai, China) and were created using the sequences given in Table 1. The relative expression levels of the targeted genes were analyzed using the 2^−ΔΔCt^ method, as previously described [35]. As an internal control, glyceraldehyde 3-phosphate dehydrogenase (GAPDH) was employed in this assay.

### 2.10. Western-Blot Assay

Briefly, the cells (2 × 10^6^ cells/well) were seeded in 6-well plates and treated for 24 h with or without the three polysaccharide samples (a dose of 40 µg/mL). After removing the culture medium, the cells were washed three times with ice-PBS (0.01 mol/L, pH 7.2) before being lysed on ice for 30 min in 100 µL of RIPA lysis buffer containing 1 mmol/L PMSF. The total protein was then obtained by centrifuging the mixture at 14,000× *g* for 5 min. The total protein content of the samples was evaluated using the BCA protein analysis kit.

The equivalent samples, which contained 15 μg of total protein, went through 12% SDS-PAGE before being transferred to nitrocellulose membranes. The membranes were incubated with the peroxidase-conjugated secondary antibody (1:5000 dilution) at 37 °C for an additional 2 h after being blocked with 5% skim milk at 37 °C for 2 h. After that, the primary antibodies (1:1000 dilution) were incubated at 4 °C for 12 h. The immunolabeled proteins were detected with enhanced chemiluminescence tools, while the Image J software (National Institutes of Health, Bethesda, MD, USA) was used to analyze the protein bands quantitatively. GADPH was also used as an internal reference in this assay.

### 2.11. Statistical Analysis

Three independent experiments or assays were used to collect all of the experimental results, which were then reported as mean values or mean values ± standard deviations. Statistical analyses (one-way analysis of variance, ANOVA) and Duncan multi-interval tests were performed using the IBM Statistical Products and Services Solutions (SPSS) software version 22 (SPSS Inc., Chicago, IL, USA). Statistics were judged to be significant at *p* < 0.05. The figure data were reported using MS EXCEL version 2003 (Microsoft Corporation, Redmond, WA, USA).

## 3. Results

### 3.1. Several Chemical Features of the Prepared Polysaccharide Samples

In this study, it was detected that the prepared YP had total saccharide and Se contents of 904.6 g and 35 mg per kilogram (on a dry basis), respectively. After the performed chemical selenylation, the resultant YPSe-I and YPSe-II had significantly enhanced Se contents of 795 and 1480 mg/kg, respectively. These results indicated clearly that the non-metal element Se was effectively conjugated to the molecules of YP during the chemical selenylation, while a higher Na_2_SeO_3_ usage (i.e., 100 mg) in the reaction system led to a higher Se conjugation (or higher selenylation extent) for YPSe-II. In total, YP, YPSe-I, and YPSe-II had different Se contents, suggesting that the three polysaccharide samples might have different abilities in the cells to exert these assessed barrier-related activities.

### 3.2. Cytotoxicity of the Polysaccharide Samples and Na_2_SeO_3_ to ICE-6 Cells

After an exposure to the three polysaccharide samples (YP, YPSe-I, and YPSe-II) for 12 and 24 h at dosages of 5–80 µg/mL, the measured viability values of the treated cells are shown in Figure 1. In total, all the assessed viability values were higher than 100%, suggesting that none of the samples had a toxic effect on the cells. To be more specific, using a cell treatment of 12 h, the cells exposed to YP, YPSe-I, and YPSe-II showed viability values of 104.8–118.5%, 104.8–139.2%, and 110.5–144.6%, respectively. Using another cell treatment of 24 h, the cells exposed to YP, YPSe-I, and YPSe-II had respective viability values of 102.0–115.9%, 103.4–120.3%, and 104.7–130.4%. Thus, these samples were regarded to have an ability to promote cell growth. The data comparison also revealed that YPSe-II and YP had the highest and lowest ability in the cells, respectively, to enhance viability values, demonstrating that the chemical selenylation of YP brought about an activity increase for the selenylated products, while a higher selenylation extent made a contribution to activity enhancement.

When the cells were treated with Na_2_SeO_3_ at the doses of 0.013–1.3 μg/mL for 12 or 24 h, the measured viability values were all less than 100% (Figure 1C). When treating the cells with Na_2_SeO_3_ at a dose of 0.013 μg/mL for 12–24 h, the measured viability values were 89.9–96.1%. However, when 40 μg/mL YPSe-II (equaling to Na_2_SeO_3_ dose of 0.013 μg/mL) was used to treat the cells for the same time periods, the resultant viability values were 122.4–131.6% (Figure 1A,B). Furthermore, an increased Na_2_SeO_3_ dose led to greater cytotoxicity on the cells; for example, the 0.13 μg/mL Na_2_SeO_3_ dose brought about viability values of 72.6–81.9%, while the 1.3 μg/mL Na_2_SeO_3_ dose led to cell death (the detectable viability value close to 0%). This fact meant that the assessed Na_2_SeO_3_, as one of the inorganic Se compounds, was very toxic to the cells, while the performed chemical selenylation of YP alleviated Na_2_SeO_3_ cytotoxicity efficiently and simultaneously endowed the selenylated products with a higher activity in the cells.

### 3.3. TEER Enhancement of the Cells in Response to the Polysaccharide Samples

To determine whether the three polysaccharide samples, YP, YPSe-I, and YPSe-II, had an ability to regulate the monolayer permeability of the cells, the TEER values of the treated cell monolayers were measured. The results showed that all samples at 5–40 μg/mL could indeed increase the TEER values of the treated cell monolayers, because the treated cell monolayers showed an increasing tendency as the treatment time was prolonged from 12 to 24 h (Figure 2). When the control cells were designed with a relative TEER value of 100%, a cell treatment of 12 h by YP, YPSe-I, and YPSe-II caused respective TEER values of 100.5–113.2%, 109.4–123.4%, and 114.6–127.3%, while that of 24 h by the three samples led to respective TEER values of 104.7–121.7%, 114.8–129.1%, and 119.6–134.1%. In other words, the treated cells had enhanced barrier integrity. The data comparison also suggested that YPSe-I and, especially, YPSe-II had a higher ability in the cells than YP to enhance the TEER value. It is thus concluded that the performed covalent Se conjugation endowed YPSe-I and YPSe-II with a higher activity to enhance the barrier function of the treated IEC-6 cells, while a higher selenylation extent caused further activity enhancement for YPSe-II.

### 3.4. Paracellular Permeability Reduction of the Cells in Response to the Polysaccharide Samples

To support the barrier promotion of the three polysaccharide samples in the cells, the cumulative transfer of FD-4 and FS-Na in the cells was also evaluated to reflect the possible changes of the targeted cell monolayers in the critical paracellular permeability. It was observed that the treated cell monolayers consistently showed a reduction in paracellular permeability (Figure 3), demonstrating that the treated cells had an enhancement in barrier integrity.

When using the FD-4 diffusion assay, the FD-4 cumulative transfer value for the control cells was designed as 100%. When treating the cells with the samples for 12 h, YP, YPSe-I, and YPSe-II at the doses of 5–40 μg/mL caused FD-4 cumulative transfer values of 79.8–91.5%, 76.5–87.3%, and 74.3–80.2%, respectively (Figure 3A). When using 24 h of treatment time, the respective FD-4 cumulative transfer values were 73.2–87.8%, 68.9–81.2%, and 62.8–78.6%, respectively (Figure 3B). Clearly, all treated cell monolayers had reduced cumulative transfer values (or increased barrier integrity). The data comparison also showed that YP and YPSe-II in the cells caused the lowest and highest FD-4 cumulative transfer values, indicating that they had the lowest and highest potential to increase barrier integrity.

Regarding the assaying results of FS-Na cumulative transport, the cell monolayers treated with the samples at the doses of 5–40 μg/mL also had reduced values of FS-Na cumulative transport compared with the control cells (FS-Na cumulative transfer value of 100%) (Figure 3C,D). Specifically, when YP, YPSe-I, and YPSe-II were used to treat the cells for 12 h, the respective values were reduced to 67.3–95.8%, 53.2–81.4%, and 44.3–69.4%. Following a 24 h exposure of the cells to the three samples, the respective values were reduced to 58.1–94.1%, 47.1–75.7%, and 32.4–60.3%. All treated cell monolayers consistently showed decreased paracellular permeability or enhanced barrier integrity compared to the control cell monolayer. The data comparison also revealed that YP and YPSe-II had the lowest and highest capacity to promote barrier integrity, while a longer treatment time also led to significant promotion in barrier integrity. Collectively, all the results suggested that the used selenylation and a higher selenylation extent endowed YPSe-I and, especially, YPSe-II with an enhanced function in the cells to enhance cell barrier integrity.

### 3.5. Anti-Bacterial Activity Upgrade and Bacterial Translocation Decrease of the Cell Monolayer in Response to the Polysaccharide Samples

To determine whether the assessed substances in the cells had an effect on the anti-bacterial activity, the three samples at 5–40 µg/mL were incubated with the cell monolayers. The supernatants from the basolateral compartments were then detected for *E. coli* counts. Briefly, the supernatants from these treated cell monolayers (treatment time of 12 h) were detected with bacterial counts reduced by 6.4–15.5% (YP), 12.0–16.1% (YPSe-I), and 13.6–16.3% (YPSe-II) compared with that from the control cell monolayer (Figure 4A). When the cell monolayers were treated with the polysaccharide samples for 24 h, the respective supernatants also showed reduced bacterial counts of 3.4–12.7%, 9.1–13.7%, and 10.5–16.3% (Figure 4B). These results suggested that all the treated cell monolayers had better barrier function than the control, reflected by the decreased numbers of *E. coli* in the basolateral compartment (i.e., a higher anti-bacterial activity). The cell exposure with YPSe-II and YP caused the highest and lowest anti-bacterial activity, respectively; meanwhile, a higher dose level of the three polysaccharide samples also showed a trend to enhance the anti-bacterial activity of the treated cell monolayers. Thus, the used chemical selenylation of YP and a higher selenylation extent could endow the two selenylated products with a higher ability to enhance the anti-bacterial activity of the targeted cell monolayer.

The bacterial translocation assay can also reflect the barrier function of the cells by detecting the presence of bacteria in the culture medium taken from the basolateral compartments. The outcomes showed that the cell monolayers treated with the three samples had an improved barrier to block off the passage of *E. coli*, because these samples always led to decreased numbers of the positive inserts (Figure 5). In detail, *E. coli* was mostly detected in the basolateral compartment after 2 h of the co-incubation of *E. coli* with the cells. In contrast to the control cell monolayer, fewer positive inserts were found in these cell monolayers treated with the three samples at 40 µg/mL for 12 or 24 h. Although longer co-incubation times (3 or 4 h) of *E. coli* with the cells caused more positive inserts (or enhanced bacterial translocation), it was generally found that YP and YPSe-II had the lowest and highest ability to slow down the bacterial translocation (or to enhance cell barrier function). The present outcomes thus confirmed once more that the chemical selenylation of YP successfully endowed the selenylated products with a greater capacity to improve the barrier function of the treated cells.

### 3.6. Promoted Intercellular Distribution of F-Actin in Response to the Polysaccharide Samples

F-actin is the most important protein constituting the cytoskeleton. If the distribution of F-actin in the cytoskeleton changes, it can cause a change in the permeability of IECs. Thus, the distribution and expression of F-actin in the cytoskeleton of the IEC-6 cells were also assessed (Figure 6). The results showed that the F-actin was localized in the intercellular junction and dispersed across the cell membrane. However, compared with the control cells, the treated cells were observed with an enhanced intensity in the fluorescence signals of F-actin, indicating a promoted distribution of F-actin between the cells. A prolonged treatment time (24 h) of the cells with the three samples led to further increased F-actin secretion (Figure 6A versus Figure 6B), while YPSe-I and, especially, YPSe-II were totally observed to cause an increased F-actin secretion compared to YP. That is, the conducted chemical selenylation and a higher selenylation extent contributed to the enhanced function of the two selenylated products, endowing them with a higher ability to promote F-actin secretion and thus enhance barrier integrity.

### 3.7. Expression Changes of the TJ-Related Genes and Proteins in the Cells in Response to the Samples

The mRNA expression levels of the three TJ-related genes ZO-1, claudin-1, and occludin were measured to further support the altered barrier function of the cells exposed to YP, YPSe-I, and YPSe-II (40 µg/mL, 24 h) (Figure 7A). The treated cells all had an increase in ZO-1, claudin-1, and occludin expression and respectively obtained 1.3–1.7-, 1.2–1.4-, and 1.5–1.8-fold up-regulation. Additionally, YPSe-I and, particularly, YPSe-II were more effective than YP in regulating the expression of the three genes. Meanwhile, the samples could also enhance the protein expression of ZO-1, claudin-1, and occludin in the treated cells (Figure 7B). Compared with the control cells, the treated cells up-regulated ZO-1, claudin-1, and occludin by 1.29–1.81-, 1.24–1.67-, and 1.27–1.89-fold, respectively. Consistent with the results about mRNA expression, it was also found that YPSe-I and YPSe-II were more effective than YP in up-regulating the expression levels of these TJ proteins. According to these results, the used chemical selenylation and the resultant higher selenylation extent gave the selenylated products a higher ability to encourage the production of the three TJ proteins, resulting in an improved barrier function in the cells.

### 3.8. Effects of the Polysaccharide Samples on the mRNA and Protein Expression of the ROCK/RhoA Pathway

The ROCK/RhoA pathway is a traditional signaling route that is important for maintaining the integrity of the intestinal barrier. Thus, we evaluated the mRNA and protein levels of RhoA and ROCK in cells treated with YP, YPSe-I, and YPSe-II at 40 µg/mL for 24 h (Figure 8). The mRNA expression of RhoA and ROCK in the cells treated with these polysaccharide samples was reduced by 0.74–0.39- and 0.61–0.85-fold, respectively, compared with that of the control cells (Figure 8A). Clearly, YPSe-II (or YP) showed the highest (or lowest) ability in the cells to down-regulate the mRNA expression of ROCK and RhoA. Meanwhile, these samples could down-regulate the protein expression of RhoA and ROCK by 0.19–0.43- and 0.60–0.75-fold, respectively (Figure 8B). It was also verified that YPSe-II and YP had the highest and lowest capacity in the cells to down-regulate the protein expression of ROCK and RhoA, respectively. These findings consistently suggested that the three samples might inactivate the RhoA/ROCK signaling pathway, resulting in an enhanced barrier function for the treated cells; moreover, a chemical selenylation (especially in a higher selenylation extent) could endow the two selenylated products (especially YPSe-II) with an enhanced ability to interfere with this signaling pathway.

## 4. Discussion

The regular operation of the small intestine is essential to the body [36]. The intestinal barrier function strictly controls the exchange of substances between the host and the external environment and serves as a filtering device for the selective passage [37]. Intestinal barrier function allows the needed nutrients to pass from the intestinal lumen into the internal environment and circulatory system, and it simultaneously prevents the invasion of harmful and toxic substances into the body [38]. The disruption of the intestinal barrier function might lead to the uncontrolled trans-epithelial flow of antigens, which in turn disrupts the immune system of the susceptible individuals, affects the host-microbial balance, and initiates an inflammatory response in the intestine or more distant tissues and organ systems, leading to the development of various diseases such as celiac disease, inflammatory bowel disease, and others [39,40,41,42,43,44]. It is now regarded that food components are able to maintain and improve the critical barrier function of the intestine or intestinal epithelial cells. For example, the polysaccharides isolated from Lycium barbarum could alleviate the inflammatory response of TNF-α in Caco-2 cells and thus combat against the dysfunction of the intestinal barrier by inhibiting the NF-κB-mediated MLC signaling pathway; subsequently, cell permeability was reduced from 170% to 130% [45]. Proteins/peptides/amino acids also can maintain intestinal barrier function. It was verified that glycine could alleviate the dysfunction of the intestinal barrier caused by Brefeldin A-induced endoplasmic reticulum stress, resulting in higher TEER values and the up-regulated expression of TJ proteins to restore the damaged intestinal barrier [46]; meanwhile, bovine lactoferrin was capable of alleviating the DSS-induced expression decrease in TJ proteins and the structural disruption of the rat gut, ameliorating the intestinal inflammation and mucosal barrier changes via the NF-κB/NLRP3 signaling pathway [47]. Polyphenols, widely distributed in plant foods, are also bioactive substances with a capacity to improve intestinal barrier function. Resveratrol has been clarified to increase the TRRE value of the Caco-2 cell monolayer and promote the secretion of two TJ proteins, namely, ZO-2 and occludin [48], while both galangal and kaempferol showed an ability to increase anti-bacterial activity and decrease bacterial translocation after their action on the IEC-6 cell monolayer [49]. Sharing the same conclusion with these studies, this study found that all of the assessed polysaccharide samples were capable of increasing the TEER value and anti-bacterial activity, reducing the FD-4/FS-Na cumulative transfer and bacterial translocation, promoting TJ protein production, and activating the RhoA/ROCK signaling pathway. Thus, they were regarded as being able to enhance the barrier function of the cells. Polysaccharides are bioactive to cells. For example, the lettuces polysaccharides could improve the phagocytosis of the RAW264.7 cells, because NO production was increased from 16 to 28 μmol/L [50]. It was found that the *Phellinus linteus* polysaccharides could inhibit the growth of thyroid cancer cells, promote cell apoptosis, and suppress cell migration and invasion [51]. Additionally, the homo-polysaccharide isolated from the dried *Cornus* pulp could reduce fasting blood glucose levels in streptozotocin-induced diabetic rats [52]. It is well known that chemical modifications such as acetylation, sulfonation, selenylation, and phosphorylation can endow natural polysaccharides with a higher bioactivity. It was thus revealed that the acetylated *Cyclocarya paliurus* polysaccharides possessed a higher anti-oxidation ability to scavenge ABTS free radicals effectively and could protect the dendritic cells from H_2_O_2_-induced damage [53], while the sulfonated kombucha polysaccharides could increase the diversity of the mouse intestinal microflora and reduce colonic fibrosis to repair the DSS-damaged intestinal barrier in mice [54]. Moreover, the selenylated polysaccharides from the *Catathelasma ventricosum* mycelium were able to reduce the blood glucose level of the streptozotocin-induced diabetic mice more effectively than the counterparts, because the former decreased the blood glucose level from 19.5 to 8.7 mmol/L, whilst the latter only reduced the blood glucose level from 19.8 to 15.8 mmol/L [55]. These published results thus proved that YPSe-I and YPSe-II were more active in the cells than YP in enhancing cell barrier integrity. This investigated chemical modification is thus regarded to have application potential to alter polysaccharide bioactivity in the intestine efficiently.

TJs are dynamic multi-protein complexes that function as selective/semi-permeable paracellular barriers. TJ proteins are bound to the cytoskeleton, forming a stable linkage system and jointly regulating the intracellular and extracellular signal transduction pathways to control cell permeability [56]. In the cytoskeleton, the intercellular microfilaments are mainly composed of F-actin, where the filamentous stress fibers in F-actin can cross the cytoplasm to form short fibers that extend into the lamellipodia of the cells to perform their corresponding functions. When the cytoskeleton is damaged, the supporting force of the intracellular skeleton is insufficient while the cells collapse, resulting in an increase in the cell gap as well as a dysfunctional structure and TJ protein expression, which in turn disrupts the cell barrier function. When ZO-1 binds to occludin to form a transmembrane protein and then connects to the cytoskeleton, it affects actin contraction and thus affects the assembly of TJ proteins, ultimately regulating cell permeability [57]. It is interesting to note that a variety of food ingredients can affect intestinal barrier function by controlling the production of F-actin and TJ proteins. For instance, the polysaccharides from *Lepidium meyenii* could enhance the barrier function of Caco-2 cells by increasing the expression of F-actin and ZO-1 [58], while chlorogenic acid could increase the expression of F-actin and three TJ proteins including ZO-1, occludin, and claudin-1 to relieve intestinal barrier injury in model mice [59]. Furthermore, α-lactalbumin peptides could enhance the intestinal barrier by regulating the expression of F-actin and ZO-1 [60].

It is crucial to learn more about the TJ-related molecular pathways in order to comprehend the impact of these polysaccharide samples on the barrier integrity of the IEC-6 cell. One of the established routes involved in cell barrier function is the RhoA/ROCK signaling system. Rutin, a naturally occurring flavonoid substance, was discovered to stop hyperglycemia from causing renal endothelial cell barrier failure by blocking off the ROCK/RhoA signaling pathway [61]. It was reported that the Astragalus mongholicus polysaccharide was able to inhibit the ROCK/RhoA signaling pathway to reduce endothelial cell permeability [62]. A large number of study results thus demonstrated that the NF-κB signaling pathway plays an important role in the occurrence and development of the intestinal barrier [63]. For example, the *Hippophae rhamnoides* polysaccharides could decrease the expression of NF-κB, TLR4, and MyD88 by inhibiting the TLR4/NF-κB signaling pathway and thus reduce barrier damage induced by inflammatory factors [64], while chlorogenic acid could enhance intestinal barrier function by inhibiting the NF-κB signaling pathway [65]. Although the Se-enriched polysaccharides such as *Agaricus blazei Murrill* polysaccharides and tea polysaccharides have been proven to enhance intestinal barrier function [66,67], it is still not verified which signaling pathway is involved in the mentioned barrier promotion. In this study, ROCK/RhoA expression was down-regulated in the cells exposed to the polysaccharide samples. It can be concluded that these samples enhanced cell barrier function through an inactivation of the RhoA/ROCK signaling pathway. However, whether these samples interfered with other signaling pathways was not identified in this study and should be clarified in future.

## 5. Conclusions

In this study, the investigated chemical selenylation of YP endowed the two selenylated products YPSe-I and, especially, YPSe-II with higher bioactivity regarding their ability to promote the barrier function of the targeted IEC-6 cells. To be more specific, for this reason, the exposed IEC-6 cells possessed improved TEER, stronger anti-bacterial activity against *E. coli*, decreased paracellular permeability, reduced bacterial translocation, enhanced intercellular F-actin distribution and TJ protein expression, and down-regulated ROCK and RhoA expression. The results thus highlighted that a covalent conjugation of minor element Se to YP can cause an activity increase in the intestine, while the exposed intestinal epithelial cells thus possess promoted barrier integrity. Overall, the covalent conjugation of Se with natural food components might endow them with enhanced biofunction in the body and should thus be given special attention in future studies.

## Figures and Tables

**Figure 1 nutrients-14-03950-f001:**
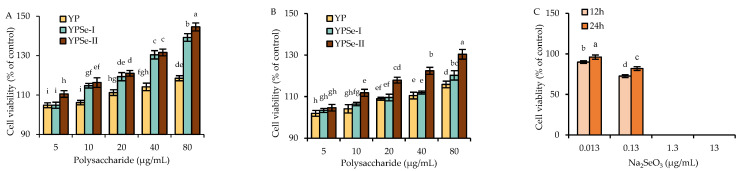
Cell viability (%) of the IEC-6 cell exposed to YP, YPSe-I, and YPSe-II for 12 (**A**) and 24 (**B**) h or exposed to Na_2_SeO_3_ (**C**). Different lowercase letters above the columns denote a significant difference in the one-way ANOVA means (*p* < 0.05).

**Figure 2 nutrients-14-03950-f002:**
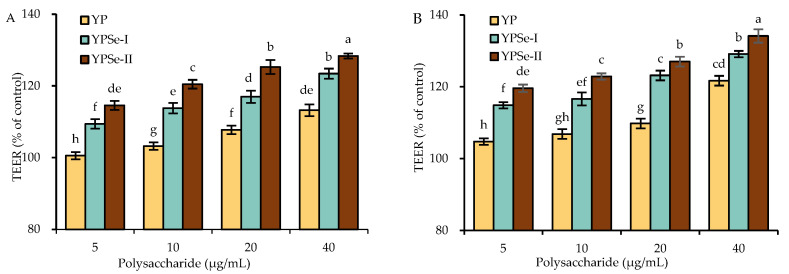
Time-responses of the TEER of IEC-6 cells treated with YP, YPSe-I, and YPSe-II for 12 (**A**) and 24 h (**B**). Different lowercase letters above the columns denote a significant difference in the one-way ANOVA means (*p* < 0.05).

**Figure 3 nutrients-14-03950-f003:**
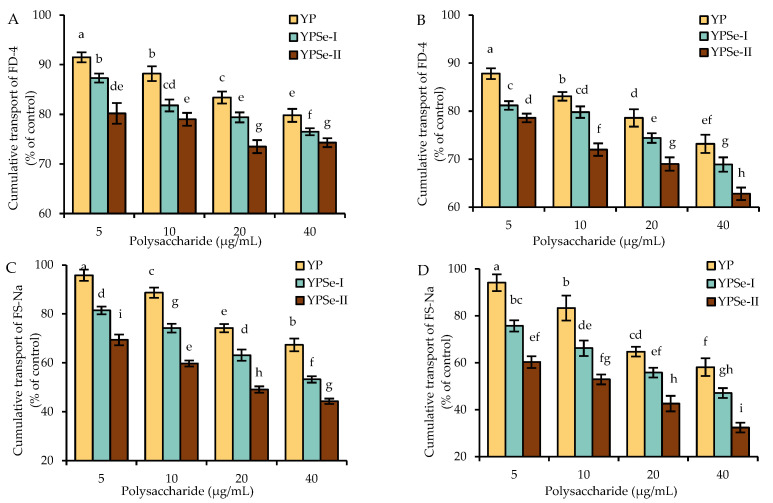
Diffusion of FD-4 and FS-Na in IEC-6 cells treated with YP, YPSe-I, and YPSe-II for 12 (**A**,**C**) or 24 h (**B**,**D**). Different lowercase letters above the columns denote a significant difference in the one-way ANOVA means (*p* < 0.05).

**Figure 4 nutrients-14-03950-f004:**
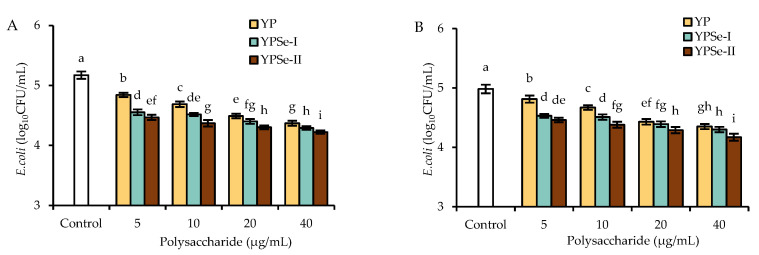
Anti-bacterial activity of the culture supernatants of IEC-6 cells exposed to YP, YPSe-I, and YPSe-II for 12 (**A**) or 24 h (**B**). Different lowercase letters above the columns denote a significant difference in the one-way ANOVA means (*p* < 0.05).

**Figure 5 nutrients-14-03950-f005:**
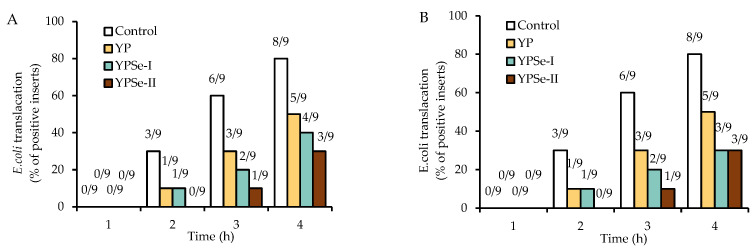
Effect of cell treatment with YP, YPSe-I, and YPSe-II at 40 μg/mL for 12 (**A**) or 24 h (**B**) on *E. coli* translocation across IEC-6 cell monolayers. The first number above the columns indicates the *E. coli*-positive inserts, while the later one above the columns indicates the assessed inserts.

**Figure 6 nutrients-14-03950-f006:**
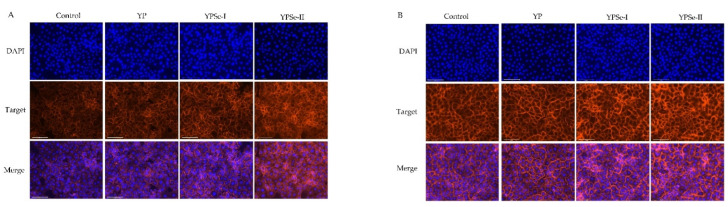
Distribution and localization of the cytoskeletal F-actin in IEC-6 cells treated with YP, YPSe-I, and YPSe-II for 12 (**A**) and 24 h (**B**). The labeled bar is 125 μm for the original pictures.

**Figure 7 nutrients-14-03950-f007:**
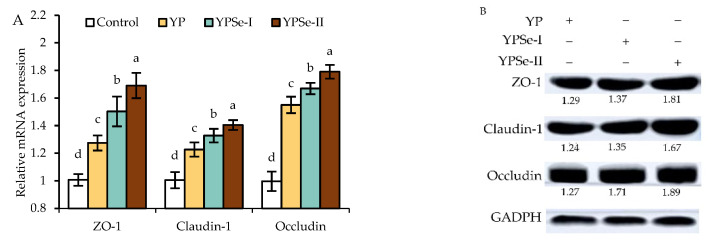
Relative mRNA (**A**) and protein expression levels (**B**) of three TJ proteins in IEC-6 cells incubated with YP, YPSe-I, and YPSe-II for 24 h. Different lowercase letters above the columns denote a significant difference in the one-way ANOVA means (*p* < 0.05).

**Figure 8 nutrients-14-03950-f008:**
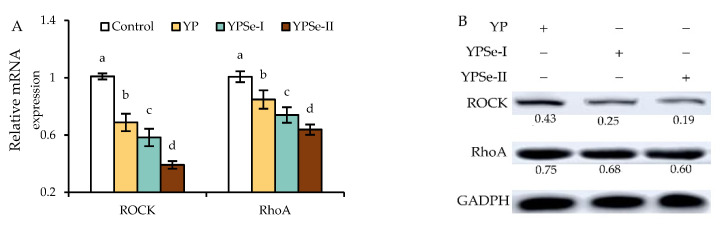
Relative mRNA content (**A**) and protein expression (**B**) of ROCK and RhoA in IEC-6 cells incubated with YP, YPSe-I, and YPSe-II for 24 h. Different lowercase letters above the columns denote a significant difference in the one-way ANOVA means (*p* < 0.05).

**Table 1 nutrients-14-03950-t001:** The sequences of the primers used in the RT-qPCR assays.

Genes	Species	Primer Sequences (5′–3′)
ZO-1	Rat	Forward: CCACCTCGCACGTATCACAAGC
		Reverse: GGCAATGACACTCCTTCGTCTCTG
Occludin	Rat	Forward: CCTCCTTACAGGCCGGATGA
		Reverse: AGCATTGGTCGAACGTGCAT
Claudin-1	Rat	Forward: GTTTCATCCTGGCTTCGCTG
		Reverse: AGCAGTCACGATGTTGTCCC
RhoA	Rat	Forward: AGGCGGGAGTTAGCCAAAAT
		Reverse: GTACCCAAAAGCGCCAATCC
ROCK-1	Rat	Forward: GGTGATGGAGTACATGCCAGGTG
		Reverse: ATCCAGTGCAAGCACGACTTCAG
GAPDH	Rat	Forward: CCCTCTGGAAAGCTGTGG
		Reverse: GCTTCACCACCTTCTTGATGT

## Data Availability

All data are contained within the article.

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
