# Peer review of "The Barrier-Enhancing Function of Soluble Yam (Dioscorea opposita Thunb.) Polysaccharides in Rat Intestinal Epithelial Cells as Affected by the Covalent Se Conjugation"

_nutrients, 2022, doi:10.3390/nu14193950_

Round 1

Reviewer 1 Report

The authors use the in vitro model to demonstrate the activities of polysaccharides extracted from Yam and also polysaccharides conjugated to Se.

1. Line 136: Not understandable. 'after a concentration'?

2. English should be extensively reviewed by native speaker.

3. Fig 1-4: Different letters able the columns indicate significant differences but not telling different among which columns.

4. Fig 5: the 0/9, 6/9 number above the columns are not understandable.

Author Response

The authors use the in vitro model to demonstrate the activities of polysaccharides extracted from Yam and also polysaccharides conjugated to Se.

  1. Line 136: Not understandable. 'after a concentration'?

Reply: Thanks for your kindly comment.

We had revised this sentence; please see the line 140 in the revised manuscript.

  1. English should be extensively reviewed by native speaker.

Reply: Thanks!

This manuscript was proofed carefully by each coauthor. We have paid our best effort to correct this manuscript, aiming to ensure its writing quality.

  1. Fig 1-4: Different letters able the columns indicate significant differences but not telling different among which columns.

Reply: Thanks!

In Figures 1-4, each group of data had been repeated at least three times. When using ANOVA statistical analysis, the conclusions shown in the figure are obtained by analyzing all groups as a whole, rather than comparing different doses of the same species. The samples were subjected to statistical analysis to show the significant different among these data.

  1. Fig 5: the 0/9, 6/9 number above the columns are not understandable.

Reply: It’s a good question.

The numbers in Figure 5 indicate the significance of the number of positive wells out of the 9 inserts performed. For example, 6/9 means that 6 positive wells appeared in 9 assessed inserts. We had revised the figure caption; please see the lines 404-405 in the revised manuscript.

Thanks!

Reviewer 2 Report

The author should enhance the quality of the manuscript, such as:

1. Please add 2-3 sentences in the abstracts related to brief method

2. Keywords, please write alphabetically.

3. Please add related references about selenylated studies in previous study in the introduction section.

4. Please add the reference in 2.4. section.

5. Please suggest what active compound contributes to this mechanism. Please delete expressions that show too speculative because the author also did not show any data related to active compounds.

6. Why did the author not measure inflammation or antioxidant activities in this study?

7. Lines 540-543, is there any relation to inflammation or antioxidant activity? If yes, please add data to this manuscript.

8. Please delete a sentence starting from "Overall .... " (lines 553-555)

Author Response

The author should enhance the quality of the manuscript, such as:

  1. Please add 2-3 sentences in the abstracts related to brief method

Reply: Thanks!

In the abstract, we briefly describe the relevant content of the experiments performed (i.e. the measured indices, lines 23-28). The present abstract has 260 words, thus we prefer not writing more details about the experimental methods used. We prefer using more words to report our results and conclusion.

As you see, too many words in abstract section might be rejected by the editor.

  1. Keywords, please write alphabetically.

Reply: A good suggestion.

The keywords are listed alphabetically as you suggested; please see lines 34-35 in the revised manuscript for details.

  1. Please add related references about selenylated studies in the previous study in the introduction section.

Reply: It’s a good idea.

Our study focused on the effects of selenylated yam polysaccharides on the functional advantages of the intestinal barrier, rather than chemical selenylation modification, and therefore the polysaccharide modification is only described briefly in the introduction (please see lines 53, and 83-86).

We prefer focusing our target on the bioactivity changes, thus not describing about the used selenylation methods in the previous studies.

  1. Please add the reference in 2.4. section.

Reply: A good idea.

Two references have been cited in this subsection. Please see the revised lines 155-158 in the revised manuscript.

Thanks!

  1. Please suggest what active compound contributes to this mechanism. Please delete expressions that show too speculative because the author also did not show any data related to active compounds.

Reply: Thanks!

We had made a careful rewriting for this manuscript, and several expressions were corrected.

  1. Why did the author not measure inflammation or antioxidant activities in this study?

Reply: It is great idea!

Anti-oxidation of polysaccharides has been investigated. We also measured this bioactivity. However, we prefer reporting barrier-promotion of the selenylated polysaccharides to the readers. Moreover, anti-inflammatory activity of the polysaccharides is also our target. We hope the results might be reported as soon.

As you see, the present results are given as 8 combined figures (19 separated pictures). Thus, the results can support our aim efficiently.

Thanks!

  1. Lines 540-543, is there any relation to inflammation or antioxidant activity? If yes, please add data to this manuscript.

Reply: Thanks!

This reference work also showed that the polysaccharides could inhibit the pathway to reduce the inflammatory factor-induced barrier loss. Whether the polysaccharides could combat against the inflammation, might be out of the present topic.

However, the reviewer suggested an interesting issue for future study. Very thanks!

  1. Please delete a sentence starting from "Overall .... " (lines 553-555)

Reply: Thanks!

We found that the polysaccharide products after selenylation had a positive effect on the intestinal barrier, so we put forward a hypothetical conclusion that the products could regulate the related signaling pathway. Because several pathways involved in barrier function, thus more pathways should be assessed. Thus, we think it is unnecessary to delete this sentence.

Round 2

Reviewer 2 Report

The paper is accepted, the author have followed reviewer comments